# FPGA-Based Implementation for Real-Time Epileptic EEG Classification Using Hjorth Descriptor and KNN

**Achmad Rizal** [1,*] **, Sugondo Hadiyoso** [2] **and Ahmad Zaky Ramdani** [2]

[1] School of Electrical Engineering, Telkom University, Bandung 40257, Indonesia
[2] School of Applied Science, Telkom University, Bandung 40257, Indonesia
* Correspondence: achmadrizal@telkomuniversity.ac.id

**Abstract:** The EEG is one of the main medical instruments used by clinicians in the analysis and diagnosis of epilepsy through visual observations or computers. Visual inspection is difficult, time-consuming, and cannot be conducted in real time. Therefore, we propose a digital system for the classification of epileptic EEG in real time on a Field Programmable Gate Array (FPGA). The implemented digital system comprised a communication interface, feature extraction, and classifier model functions. The Hjorth descriptor method was used for feature extraction of activity, mobility, and complexity, with KNN was utilized as a predictor in the classification stage. The proposed system, run on a The Zynq-7000 FPGA device, can generate up to 90.74% accuracy in normal, inter-ictal, and ictal EEG classifications. FPGA devices provided classification results within 0.015 s. The total memory LUT resource used was less than 10%. This system is expected to tackle problems in visual inspection and computer processing to help detect epileptic EEG using low-cost resources while retaining high performance and real-time implementation.

**Keywords:** EEG; epileptic; digital system; FPGA; real-time

## 1. Introduction

Epilepsy is a common brain illness caused by aberrant cell activity. It typically affects more than 50 million people worldwide, most of whom live in underdeveloped nations [1]. The electroencephalography (EEG) signal is one of the tools used by doctors and neurologists to assess brain nerve activity, documented as spikes for the medical team to visualize [2]. However, the prolonged visualization of EEG records by medical teams in detecting the existence of epileptic attacks is tedious, time-consuming, and prone to human error.

Over the years, several methods have been developed to detect and classify seizures on the EEG to address the mentioned problem. From a signal domain perspective, EEG signal analysis can be divided into time, frequency, and time–frequency domains. The EEG signal does not require a transformation process in time domain signal analysis [3,4]. The characteristics often used in the time domain are statistical, such as mean, variance, skewness, kurtosis, entropy, and energy. Fourier transform is widely used to convert EEG signals from the time to the frequency domain by determining characteristics such as square frequency, mean frequency, etc. [5]. Another tool often used is Hilbert marginal spectrum analysis [6,7]. Short-time Fourier transform (STFT) has become the most popular method for transforming signals from the time domain to the time–frequency domain [8,9]. Another study simulated epilepsy detection on long-term EEG using multimodal feature extraction in the time domain including signal complexity and first and second-order statistics [10]. Feature extraction in the time domain using a signal complexity approach is also often performed on multiscale signals, as reported in the literature [11–13]. Most studies related to the simulation of classification or detection of epileptic EEG yield high accuracy. However, the various techniques developed have only been realized in the form of software.

Several studies were conducted to build a real-time EEG processing system, and one uses a field-programmable gate array (FPGA) [14]. Sundaram et al. developed median and moving average filters for pre-processing EEG signals on FPGAs (Virtex-5). The test results showed that the median filter provided the best performance regarding area, power, and delay. Wöhrle et al. used Xilinx Zynq to process EEG and EMG signals for movement prediction [15]. The research accuracy obtained from fixed-point computation is not significantly different from a PC comprising double-precision floating-point. The advantages of using FPGA are cost-effectiveness, low power, and the possibility to build an embedded system [16]. There are still many developments in implementing EEG signal processing methods on FPGA.

One application of real-time EEG signal processing is for the detection of epilepsy. Saric et al. used continuous wavelet transform (CWT) and MLP-ANN to detect epilepsy based on EEG signals [17]. The method was implemented using FFGA and tested using the Temple University Hospital Seizure Detection Corpus (TUH EEG Corpus) database. The highest accuracy reached 95.14% using the configuration (5-12-3) on the MLP-ANN. Jose et al. used an FPGA implementation of EEG epileptic detection using extreme learning (ELM) [18]. The EEG signal was separated into several bands, namely gamma, beta, alpha, theta, and delta brain rhythms, and modeled by linear prediction theory. Another method to implement epilepsy detection using FPGA is variational mode decomposition (VMD) [19]. This study used reduced deep convolutional neural networks (RDCNN) and multi-kernel random vector functional link networks (MKRVFLN) as classifiers. In the studies mentioned above, the focus of the research tends to be on classification methods with advanced feature extraction methods. Simple methods without signal transformation have not been thoroughly explored.

In this research, a real-time EEG epilepsy classification system was realized using an FPGA and a simple method known as the Hjorth descriptor, which measures the complexity of the signal in the time domain [20]. This method does not require a signal transformation process, indicating that the computational load can be reduced. For classification, we used KNN because no training process is needed to build the model as in other classifiers. KNN only requires training data to calculate the distance from the test data. Thus, the implementation process becomes more straightforward. With the implementation of the system in hardware, it is possible to realize a portable device for classifying EEG epilepsy signals in the future. The designed system is expected to have high accuracy with acceptable processing delay.

## 2. Material and Methods

### 2.1. Dataset Description

The proposed method was tested using the well-known Bonn University EEG database, which was downloaded for free at https://www.upf.edu/web/ntsa/downloads/ (accessed on 20 April 2022) [21]. This database is divided into five sections, namely A, B, C, D, and E, with each subset containing 100 EEG signals saved as a text document (.txt). Each EEG signal is 23.6 s long and comprises 4097 samples collected at a rate of 173.61 Hz. The A and B subsets represent five healthy subjects with their eyes open and closed, respectively. The C and D subsets are made up of five individuals who had fully recovered from seizure control following epileptic surgery. The EEG signals with epileptic seizure events detected in the epileptogenic zone make up subset E. The signals are normal in the A and B subsets, interictal in C and D, and ictal in E.

### 2.2. Feature Extraction

The Hjorth descriptor was proposed for measuring the time domain EEG signal's dynamic [22]. It has also been used as a marker in another biological signal, such as electromyogram (EMG), electrocardiogram (ECG) [23], and lung sound processing [24]. This method has important parameters, namely activity, mobility, and complexity. The activity measures the signal's strength, as well as the irregularity of the time function

theoretically. The activity is defined as the variance of time function $\sigma_x{}^2$, which can be described as the square of the average distance from each sample to its mean.

The mobility parameter represents the proportion of standard deviation of the power spectrum. This is defined as the division result of variance of the first derivative of the signal $x(n)$ and the variance of the signal $x(n)$. Lastly, the complexity parameter represents the change in frequency calculated by dividing the standard deviation comparison between the second and first derivative, and the first derivative and the original signal.

The Hjorth parameter equation is shown in Equations (1)–(3) [20].

$$Activity = \sigma_x{}^2 = \frac{\sum_{n=1}^{N-1}(x(n) - \overline{x})^2}{N} \tag{1}$$

$$Mobility = \frac{\sigma_x'^2}{\sigma_x{}^2} \tag{2}$$

$$Complexity = \frac{\sigma_{x''}/\sigma_{x'}}{\sigma_{x'}/\sigma_x} \tag{3}$$

### 2.3. KNN Classifier

K-Nearest Neighbor (KNN) is a classification method that uses the distance of test and training data to determine the K values [25]. The value of K is usually odd to avoid the same number of labels in the case of binary classification [26]. One of the essential components of the KNN method is distance calculation, which often uses include Euclidean, Manhattan, and Mahalanobis [27]. In this research, we implemented this calculation method using the Euclidean method.

### 2.4. System Model

This research focuses on the real-time implementation of the EEG classification system according to the algorithm described earlier. The EEG epileptic classification system is implemented on FPGA using VHDL code, which was selected based on the need for high parallel computation for the KNN classifier process, specifically for many comparison samples. Hardware architecture design has become crucial to determining system performance, specifically in supporting larger-scale development in the future, which will increase the number of calculations required. Therefore, the proposed architecture is a scalable and source-efficient system capable of accommodating further development using more comparative datasets in the KNN process. With relatively slow data, the FPGA's flexibility and parallel computation capability will play an important role in accommodating the entire process in real time.

Figure 1 shows the workflow of the designed system, which indicates the gradual process used by EEG signals to receive data digitally. The data collected will be accommodated in a series called frames, which is also determined according to the dataset size. Furthermore, the calculation process for each Hjorth parameter comprising activity, mobility, and complexity is conducted in the frame. These three features are used in the classification stage to determine query labels, categorized in three classes, namely normal, ictal, and interictal.

The implemented system is divided into two main processing parts, namely signal for feature extraction and KNN core as a classifier. The input from the system is integrated with the serial UART as the input port during the experiment verification process. The EEG signal received through the UART is then de-serialized into signed fixed-point data with a width of 16 bits. These sampling data are processed directly to the feature extraction process to obtain the Hjorth parameters, with each extracting process cycle carried out in real time for every 4096 samples. The process of controlling the speed of processed data is dependent on the sender, such as the EEG sensor or the PC. There is no control on the hardware that limits data reception according to the frequency received. However, this is not a problem, because the feature extraction performed by calculating the Hjorth

parameter is not a function related to the frequency response. Figure 2 shows the training phase on the PC and the system implementation on FPGA.

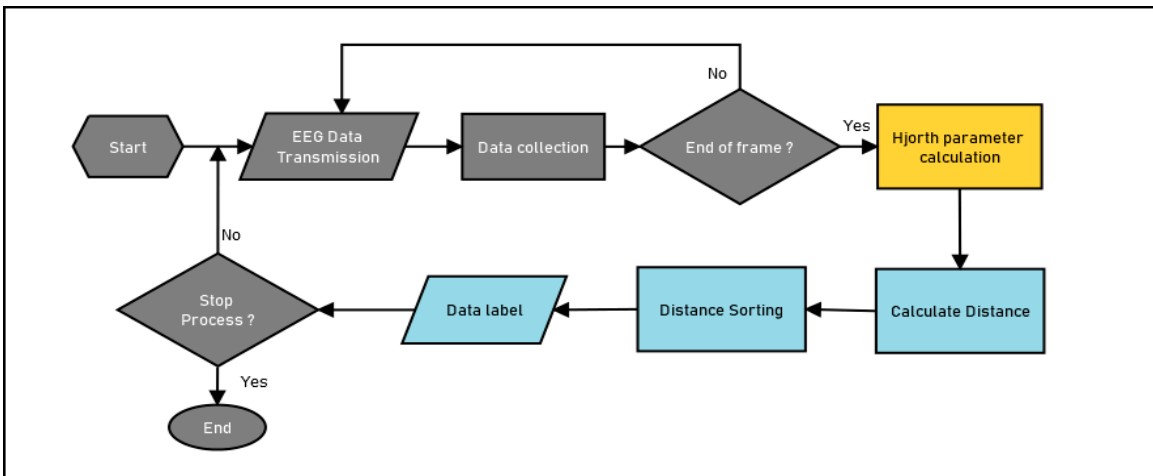

**Figure 1.** The complete system flowchart for the proposed real-time epileptic EEG classification.

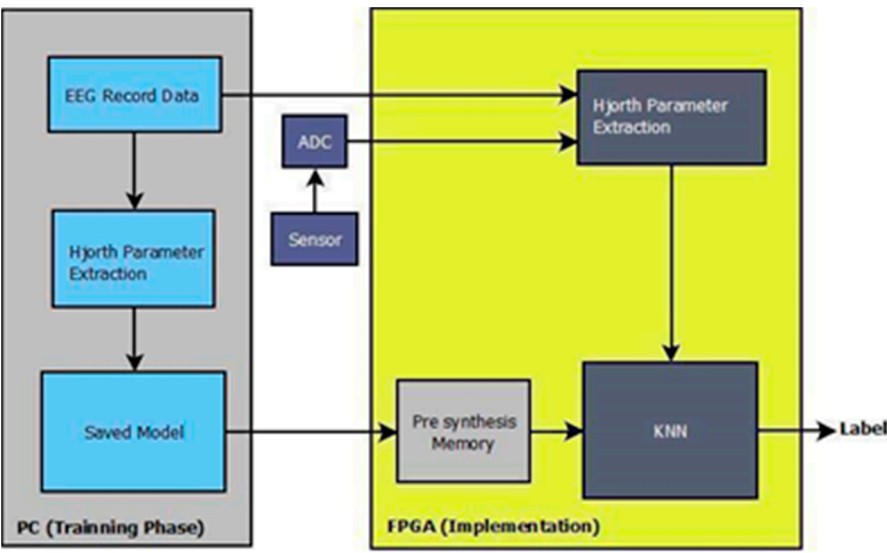

**Figure 2.** System blocks consist of three main parts: PC for the training stage, digital EEG as the alternative input provider, and FPGA as the platform for the algorithm implementation.

The three Hjorth parameters that have been described previously are the output of this feature extraction block. Furthermore, the extracted features are input from the KNN core and then compared with 192 samples from preliminary training, divided into 3 categories with 64 each. This implementation compares the results of using different K values to determine the level of accuracy of this system. The output is the classification of the system categorized into 3 kinds. Training data are stored as pre-synthesis memory. For each classification cycle, extracted features obtained from the 4096 points are compared to the stored training data using a brute force scheme. This is carried out by considering the very slow sampling frequency compared to the system clock speed and the small number of samples.

## 3. FPGA Hardware Architecture

### 3.1. Feature Extraction Datapath Architecture

The Hjorth parameter consists of activity, mobility, and complexity, calculating signal power, average signal frequency, and change in frequency, respectively. In calculating

the three parameters, Hjorth uses the variance parameter to ensure the computational complexity is low and still obtains the frequency spectrum information contained in the EEG signal. The complete equation is shown in Equations (1)–(3), and its transformation into hardware architecture is presented in Figure 3.

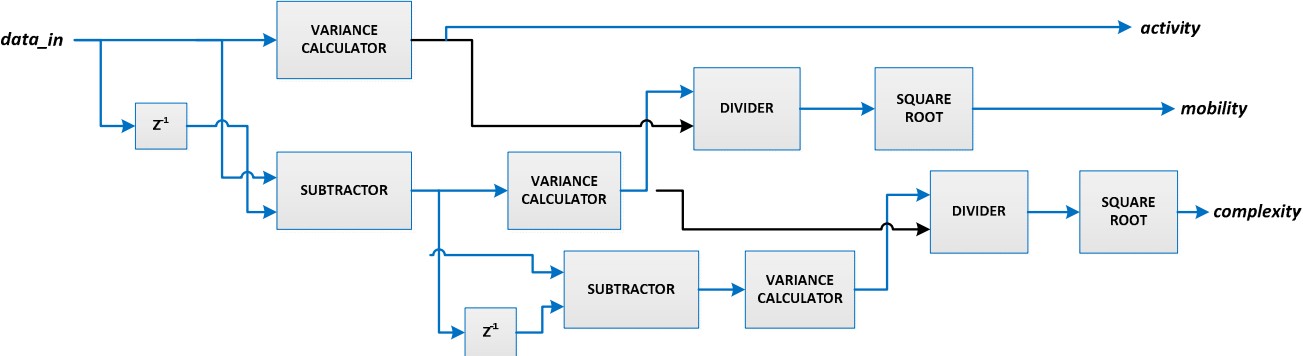

**Figure 3.** Feature extraction top level architecture to process EEG data input into three Hjorth parameters.

The equations illustrate that the main parameters of this algorithm are extracting the variance and mean values, followed by an additional division operation. Furthermore, these two parameters can be calculated by performing iterative addition of received value, followed by division operation to determine the mean. Iterative addition is also again performed for the distance of each value by squaring the subtraction of each value with the previously calculated mean. Although this approach is mathematically straightforward, it is impractical to implement in real-time systems where data run continuously and calculations can only be performed after the stream ends. It is even more complicated to implement in an FPGA because it requires memory implementation to buffer all the data after calculating the mean and re-reading to determine the variance. Therefore, to implement the algorithm, the statistical parameters need to be calculated as running mean and variance using Equation (4).

$$\sigma_x^2 = \frac{\sum_{n=1}^{N-1} x(n)^2}{N} - \frac{\left(\sum_{n=1}^{N-1} x(n)\right)^2}{N} \tag{4}$$

Signal processing applied to EEG for the feature extraction is not in the form of a filter with a certain frequency response, but it comprises differential and mean calculation; hence, no specific clock frequency is required in the implementation system. The calculation of the required parameters will focus on the amount of the incoming value for each sample, according to the sampling rate received by the system.

Logic devices such as FPGA implement the system square root using a separate algorithm developed by Putra [28]. Incoming sampling data will be given a delay in the form of a shift register to calculate the first and second data differential. Furthermore, the mean is calculated in a separate component using a total of 4096 data, divided to obtain the mean value, selected to minimize the complexity of the division process. Figure 4 shows an FPGA implementation for calculating variance.

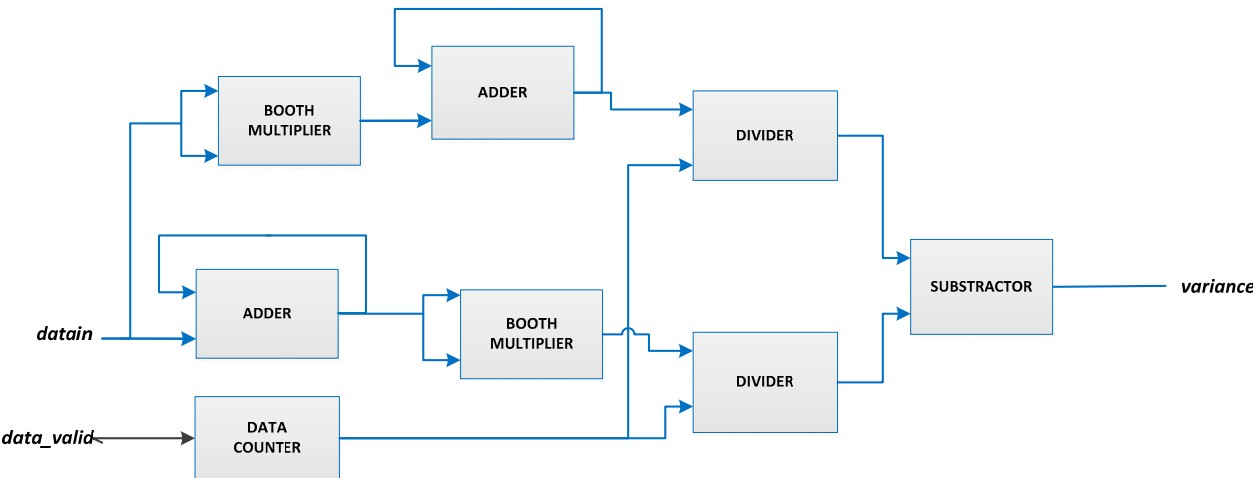

**Figure 4.** Variance calculation block implemented on the proposed system, designed to be able run on the real-time application.

### 3.2. KNN Processor Architecture

The KNN algorithm calculates the distances between the query and all the predefined labeled samples. It continues with selecting the specified number of closest examples, then labels the query to determine the most frequent label. The sample data have four defined objects, identity, activity, complexity, and mobility values, as well as classification. The first three are used for the Euclidean distance calculation, while the last is for class determination. By labeling the Hjorth parameter as $x$, $y$, and $z$, the Euclidean distance between the input signal and all training data is calculated using Equation (5). The subindex t indicates training data, while $i$ is the training set array index.

$$d(i) = \sqrt{(x - x_t(i))^2 + (y - y_t(i))^2 + (z - z_t(i))^2} \tag{5}$$

The simplest version to implement the KNN algorithm is based on the brute force mechanism, with other techniques applied to improve performance efficiency using K-D Tree [29]. This implementation is carried out using a small dataset and an FPGA which allows parallel calculations. However, for flexibility and calculation of source optimization on performance, the KNN core is still designed to be configurable, specifically for the number of PEs that work in parallel. The implementation of the distance calculation is presented in Figure 5.

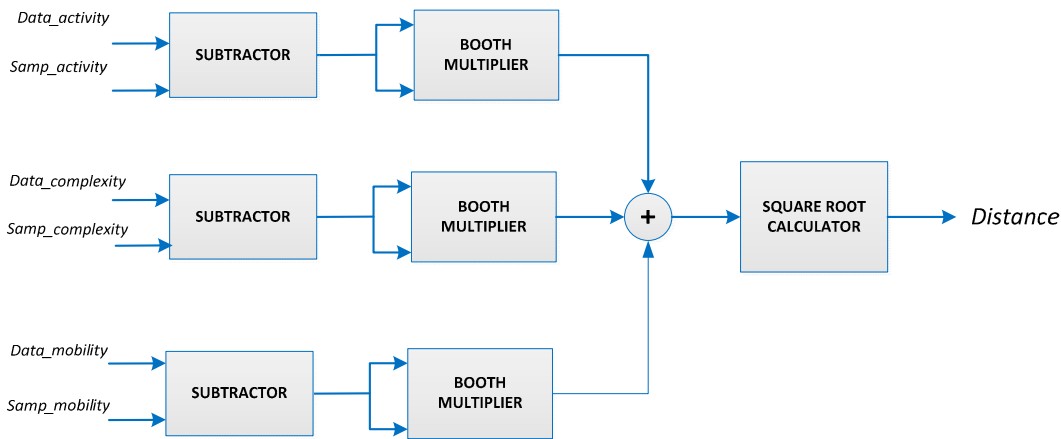

**Figure 5.** Distance calculation block used to calculate distance in KNN process; all the blocks are implemented using primitive logic to support further development on different KNN configurations.

Furthermore, after the entire distance calculation process is completed, the samples are sorted based on the closest distance. During this KNN process, each sample is seen as an object with a class identity. Sorting is applied only to nine samples, with the minimum distance sorted by identifying its class. The closest samples in the experiment session also act as a determinant according to the number of K of the KNN to be used, such as 1, 3, 5, 7, or 9. On the hardware side, the sorting process is carried out on nine samples with a minimum distance, with a difference in the samples involved for decision making. Figure 6 shows the KNN architecture implemented in FPGA.

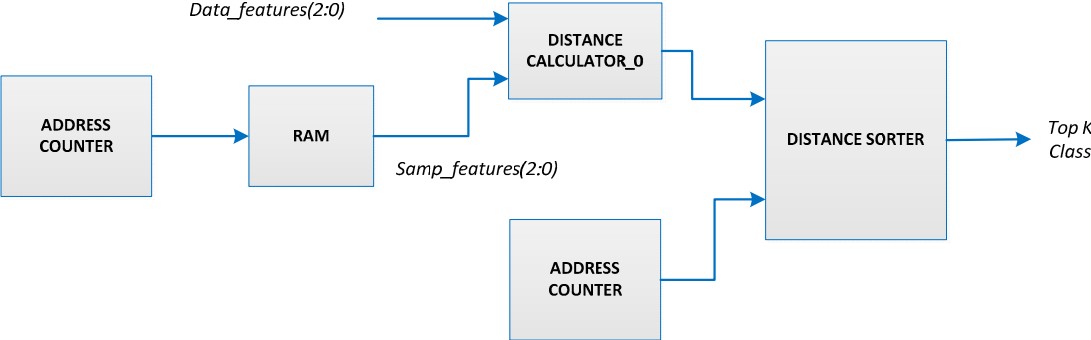

**Figure 6.** KNN top-level architecture implemented on the proposed system where the Hjorth parameters from EEG input are compared to saved data models and sorted for the maximum likelihood class.

In KNN, one factor determining the length of the process is the number of pre-calculated samples used in the distance calculator, which equals 192. This means that 192 distance calculations are needed. The more processing elements used, the less time needed to complete the function of the KNN with more logic cells. In Figure 6, an alternative implementation comprising a single processing element was used to process all 192 operations. A counter will act as a scheduler in this implementation scheme by sequentially providing an address to RAM used to calculate the sample. After calculating the distance, the process moves to the sorter stages, where the results are compared and sorted.

### 3.3. Simulation and Verification

After the described hardware architecture was translated into VHDL, the simulation was conducted using Modelsim software. This was followed by manually verifying the calculation results of each feature obtained. After each verified function implementation, simulations were conducted to perform timing analysis to determine the number of clock cycles required for each process.

A snippet of the simulation results with a clock frequency of 100 MHz is shown in Figure 7. The feature extraction process starts from the entire EEG data stream. As many as 4096 points are obtained and marked with the first cursor line. It ends at the second cursor, which requires 960 ns or 96 clock cycles. The next process is calculating the distance from all samples, completed after 10,505 cycles marked with the third cursor. This is the longest processing phase, where 192 distances were calculated, averaging 55 clock cycles to obtain distance results for each sample. After all the distances were calculated, the sorting process continued until the system produced a valid decision at 383 clock cycles. From the waveform shown in Figure 7, it can be seen that nine signals with different colors are the result of sorting from the calculated distance. This set is the nearest neighbor, which has two value components, namely the distance and the origin of the EEG class. These nine values are options used in determining the results of KNN based on the configuration of the number of K selected.

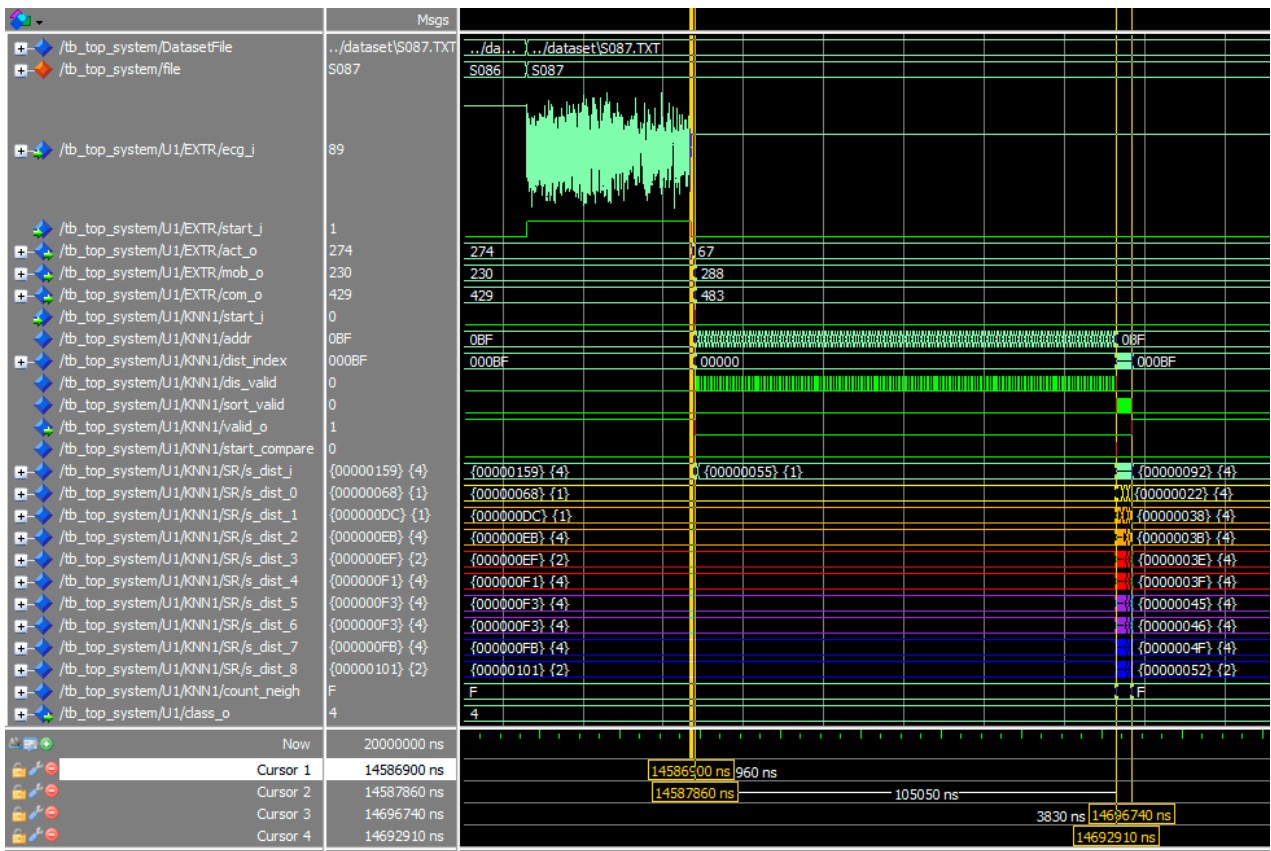

**Figure 7.** Waveform simulation results for one of the ictal samples; the rainbow waveforms show the closest saved models to the input signal where its values are distance and class (4 = ictal, 2 = normal).

## 4. FPGA Implementation

### 4.1. Experimental Setup

After defining the hardware architecture, the design was translated into VHDL code and synthesized using Xilinx Vivado 2018.2. Xilinx Zynq XC7Z030SBG485 was selected as the implementation device target. The Zynq-7000 series is a low-cost All-Programmable System-on-Chip device from Xilinx comprising two main parts, programmable logic (PL) and programmable system (PS), which are equipped ARM Cortex-A9 cores. This algorithm was implemented in the PL section, where the PS was only used to simplify the real-time testing experiment process. PS functions as an interface between the system and external devices that transmit EEG signal data, which in this experiment is a PC run periodically by a simple Python script via UART. Figure 8 shows the final block design where the experiment runs.

The initial part of this system is an interface in the form of a UART, and then the data are forwarded to the deserialization mechanism and collected with a width of 16 bits. This means that it takes two 8-bit data transfer processes through the UART to obtain one data sample. The use of UART was selected because the data from EEG are relatively very slow, and capable of providing the flexibility of data sources. In writing this paper, verification was carried out using offline data from a PC; hence, UART is considered to be an easy communication process.

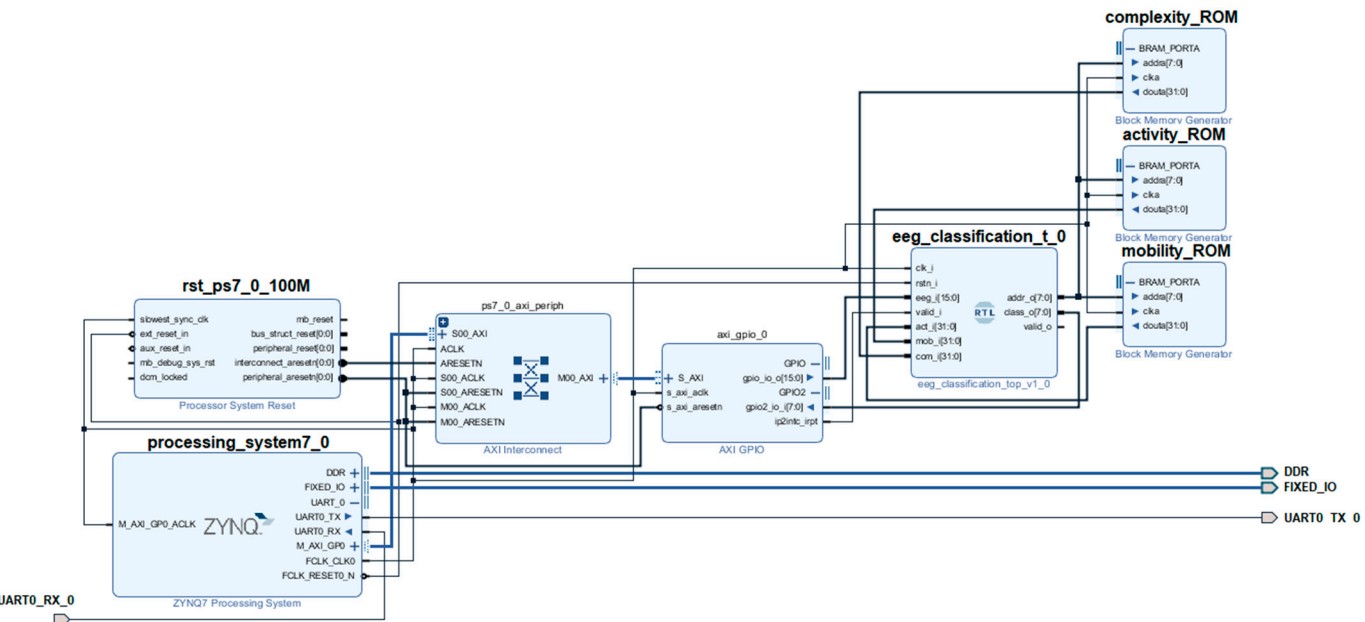

**Figure 8.** Top-level system in Vivado implementation; the proposed algorithm is integrated with other blocks such as ROM and the Zynq processor for further development.

### 4.2. Resource Consumption

In this experiment, the EEG sample data used were quantized into 12 bits with a range of $-2048$ to $2048$. The data received by PS through UART were combined for every two incoming 16 signed fixed-point bits. Furthermore, PS was used to de-serialize and issue the data to the implemented design in the PL section through GPIO, with another 1 bit as a valid data signal. For each operation performed on this proposed design, a 32-bit fixed-point system was also implemented. For comparison, in the KNN section, several variations of the number of cores used were conducted, resulting in the use of different logic resources, as shown in Figure 9.

| Name | Slice LUTs (53200) | Slice Registers (106400) | F7 Muxes (26600) | F8 Muxes (13300) | Slice (13300) | LUT as Logic (53200) | LUT as Memory (17400) | Block RAM Tile (140) | Bonded IOB (200) | Bonded IOPADs (130) | BUFGCTRL (32) |
|---|---|---|---|---|---|---|---|---|---|---|---|
| design_top_wrapper | 5202 | 10196 | 822 | 409 | 4027 | 5142 | 60 | 1.5 | 2 | 130 | 1 |
| design_top_i (design_top) | 5202 | 10196 | 822 | 409 | 4027 | 5142 | 60 | 1.5 | 0 | 0 | 1 |
| activity_ROM (design_top) | 0 | 0 | 0 | 0 | 0 | 0 | 0 | 0.5 | 0 | 0 | 0 |
| axi_gpio_0 (design_top_a) | 103 | 201 | 0 | 0 | 59 | 103 | 0 | 0 | 0 | 0 | 0 |
| complexity_ROM (design_) | 0 | 0 | 0 | 0 | 0 | 0 | 0 | 0.5 | 0 | 0 | 0 |
| eeg_classification_t_1 (de | 4705 | 9496 | 822 | 409 | 3782 | 4705 | 0 | 0 | 0 | 0 | 0 |
| inst (design_top_eeg_ | 4705 | 9496 | 822 | 409 | 3782 | 4705 | 0 | 0 | 0 | 0 | 0 |
| EXTR (design_top_) | 1395 | 1589 | 0 | 0 | 522 | 1395 | 0 | 0 | 0 | 0 | 0 |
| KNN1 (design_top_) | 3304 | 7907 | 822 | 409 | 3299 | 3304 | 0 | 0 | 0 | 0 | 0 |
| mobility_ROM (design_to) | 0 | 0 | 0 | 0 | 0 | 0 | 0 | 0.5 | 0 | 0 | 0 |
| processing_system7_0 (d | 0 | 0 | 0 | 0 | 0 | 0 | 0 | 0 | 0 | 0 | 1 |
| ps7_0_axi_periph (design | 377 | 466 | 0 | 0 | 178 | 318 | 59 | 0 | 0 | 0 | 0 |
| rst_ps7_0_100M (design_ | 17 | 33 | 0 | 0 | 12 | 16 | 1 | 0 | 0 | 0 | 0 |

**Figure 9.** Logic source used for the proposed system implementation on Xilinx Zynq XC7Z030SBG485.

The system discussed is focused on the entity named eeg_classification, which is separated again into two main parts named EXTR for the feature extraction and KNN1 for the KNN core. The total logic source required for the main block is less than 10% of the total slice and registers available, where the KNN part takes the majority of components.

## 5. Results and Discussion

### 5.1. Dataset

The dataset used to conduct this research consists of 300 signals of normal, ictal, and interictal categories, with 100 signals for each class. Furthermore, each category was randomly divided into 64 KNN train and 36 test data. There are three features extracted from each part of the sample in the form of activity, complexity, and mobility, which are calculated first through the Python script. Figure 10 shows a comparison of the calculation results between two EEGs with varying values for each feature. Samples with file names starting with S, N, and O are EEGs belonging to the interictal, ictal, and normal categories, respectively.

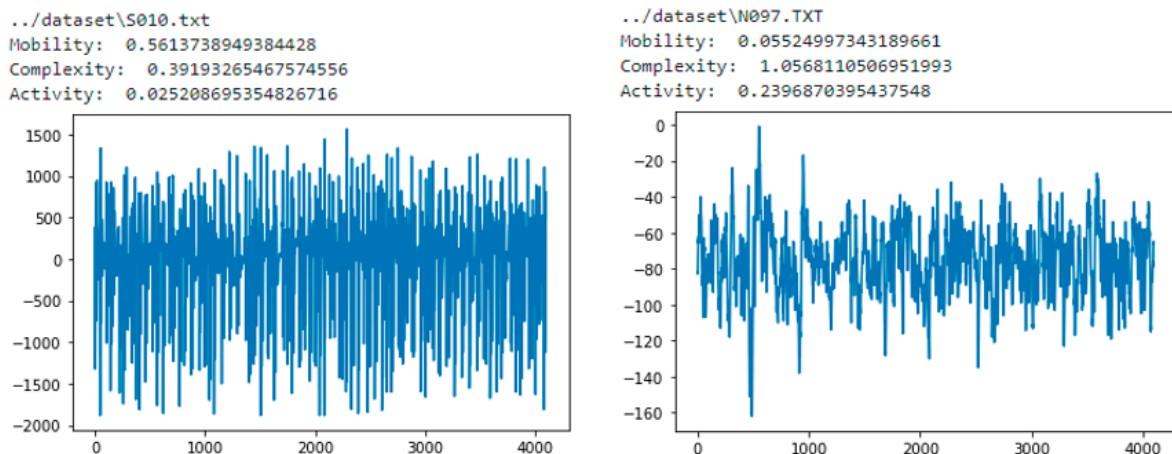

**Figure 10.** Comparison results for two sample files from two different classes, ictal and interictal. Interictal signals normally have lower mobility but higher complexity values compared to ictal.

Within Bonn University's EEG database, there are generally three data classes: normal, interictal, and ictal. In more detail, there are two measurement conditions for the normal dataset: the eyes are open (dataset Z), and the eyes are closed (dataset O). Meanwhile, in the interictal dataset, there are also two measurement methods: measurements using an intracranial electrode (dataset N) and measurements in the epileptogenic zone (set F). There is only one dataset for the ictal condition (dataset S). Because in principle, there are only three conditions, we used only three classes of data in this study. Empirically, classification results using five data classes will produce lower accuracy compared to using three data classes [30].

Figure 11 shows how the distribution of the calculation results for each feature was used to group each category. It can be seen that complexity has the most extensive range of values, around 0–1.4, while mobility and complexity have relatively more minor ranges of 0–0.6 and 0–0.1, respectively. Based on the consideration of the value range, the feature sample data are stored on RAM in a 32-bit fixed point format with a resolution of 1/1024.

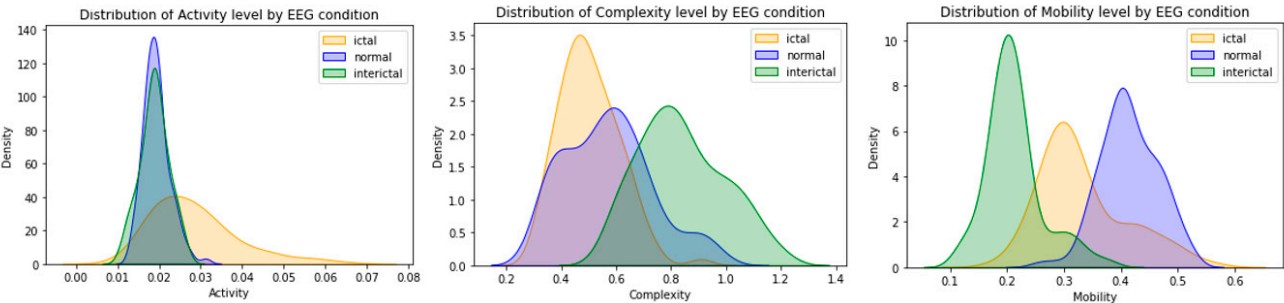

**Figure 11.** Distribution of each feature calculation result shows the different characteristics from each EEG signal class.

Performance evaluation is also conducted by comparing activity, mobility, and complexity calculation results in VHDL and Python. Table 1 shows a sample comparison of Hjorth parameters on several EEG signals.

**Table 1.** Classification results for each K.

| Configuration | Correct Result | | | | Accuracy |
|---|---|---|---|---|---|
| | Ictal | Interictal | Normal | Total | |
| K = 1 | 30 | 33 | 29 | 92 | 85.18% |
| K = 3 | 30 | 34 | 32 | 96 | 88.89% |
| K = 5 | 31 | 34 | 31 | 96 | 88.89% |
| K = 7 | 31 | 35 | 31 | 97 | 89.81% |
| K = 9 | 31 | 35 | 32 | 98 | 90.74% |

*5.2. Comparison of Hjorth Parameter Calculations between VHDL and Python*

The difference in calculation results between the Python model and algorithm implementation on VHDL is shown in Figure 12. The horizontal axis shows the error distribution values calculated using the Python model calculation as presented in Equation (6):

$$Error\ Rate = \frac{|VHDL\ result - Python\ Model|}{Python\ Model} \times 100\% \tag{6}$$

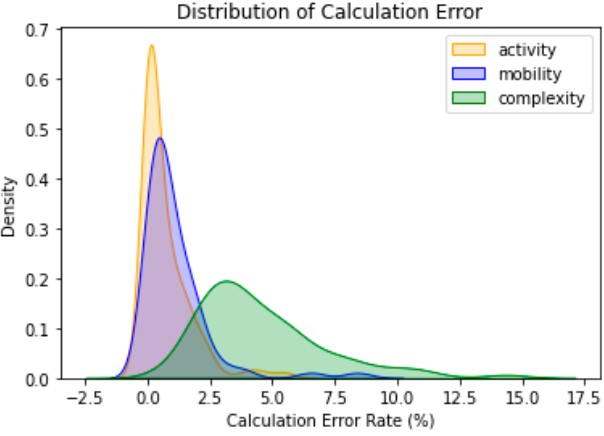

**Figure 12.** Distribution of calculation error of Hjorth parameter between VHDL and Python. Complexity value has the longest calculation path, resulting in a higher error calculation rate.

This calculation error is caused by the use of fixed points in the implementation, which has a limited range of fractional values rounded off from division and square root calculation. The activity value has the shortest calculation formula; hence, the error percentage is minimal compared to the other Hjorth parameters with an average close to 0%. This is different from the complexity, which has the longest operation, with an average calculation error of around 3%. Overall, from the three calculated features, the error value that is generated is relatively small and consistent for all features, so it is hoped that the designed system will still produce high classification accuracy.

*5.3. Classification Accuracy*

KNN is used for the classification of normal, interictal, and ictal EEG using K variations of 1, 3, 5, 7, and 9. The output length of 3 bits defined as "001", "010", and "100" represents interictal, normal, and ictal with hexadecimal values of 1, 2, and 4, respectively. Figure 13 shows a classification simulation with a normal EEG input signal. In this simulation, the number of neighbors is 5, with the final decision based on the most predicted results.

The yellow box in Figure 12 indicates misclassification, with normal detected as interictal. In the proposed system, the classification process will depend on the Hjorth parameter calculation results. In the example of the misclassification shown in Figure 13, both samples have a relatively greater complexity value compared to other samples from the normal category. The complexity parameter has the largest value range compared to the other two parameters; thus, it can be considered the most decisive parameter due to no additional parameter weight in KNN distance calculation. As the result, the samples have more neighbors coming from the interictal class rather than the normal class. Thus, the results of the pooling of the five closest neighbors show that the majority of classes are 1, interictal.

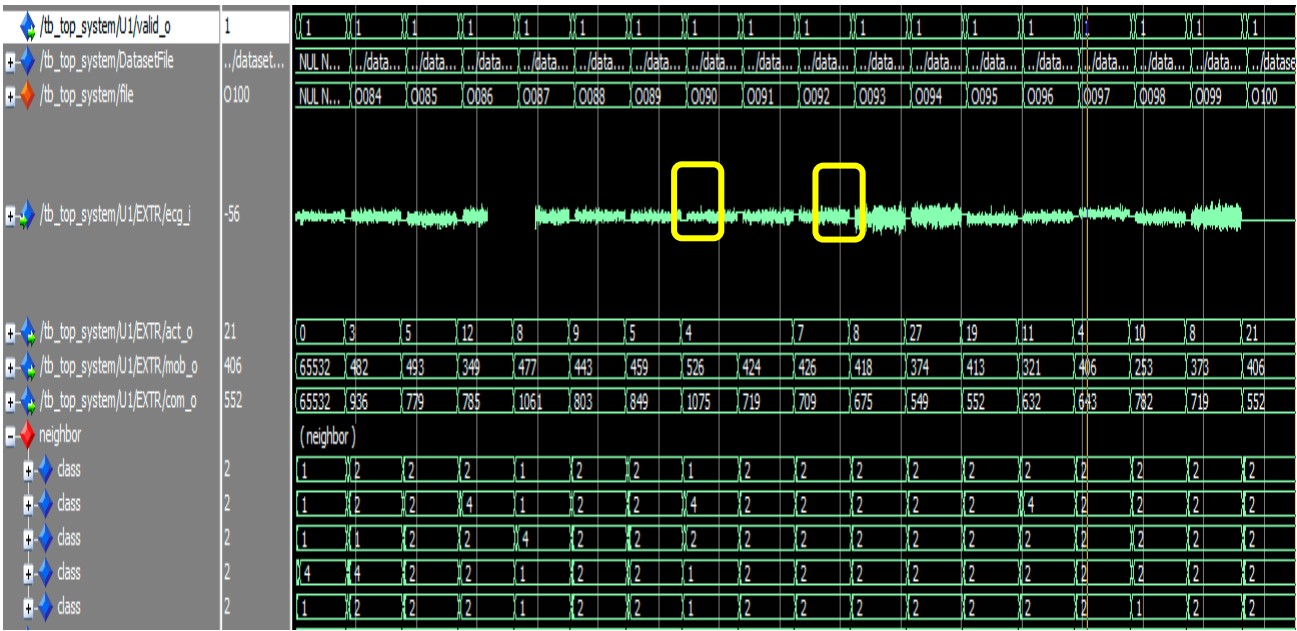

**Figure 13.** Classification simulation with normal class as input. The yellow boxes show that the inputs are wrongly classified because the closest neighbor comes from the interictal (1) class.

The overall performance evaluation of the system is carried out by dividing the test and training data with a ratio of 36:64. The total test data are 108, with 36 signals for each class. The FPGA board receives an EEG signal randomly sent over the serial interface. The classification accuracy for each K value is presented in Table 1. The number of K is linear with increasing accuracy, where the highest accuracy of 90.74% is achieved at a value of 9. Accuracy increased significantly at K > 1 compared to K = 1, but this did not occur at larger K. From the previous study, K is linear to accuracy but has an optimal value [31]. In other words, a larger K does not necessarily increase accuracy [32].

The value of K can be increased to find the optimal performance, but in this proposed system, the value of K is limited to 9. At K = 5, 7, and 9, the increase in accuracy is not higher than 1%. An enormous K value will give a lower variance but increased bias [32]. Another reason is limited memory resources with the use of a larger K.

### 5.4. Comparison with Previous Research

Table 2 presents several studies using FPGAs for the detection of epilepsy. Most research in Table 2, used datasets were from Bonn University [17–19,33] while others used datasets from Temple University Hospital [34]. For signal processing, all previous studies used a decomposition process, starting from DWT, CWT, and VMD, and the decomposition of EEG signals into the delta, theta, alpha, beta, and gamma signals. Previous research used various classifiers from SVM and ELM for deep learning. The proposed method generally produces lower accuracy compared to earlier studies listed in Table 2. However, the method used in this study is relatively simple and does not require decomposition or

signal transformation, with a classifier that does not require a separate training process. It is hoped that the results of this study can be used for real-time detection of epilepsy in research on EEG signals.

**Table 2.** Comparison with previous research.

| Reference | Method | Classifier | Dataset | Result |
|---|---|---|---|---|
| Meddah et al., 2020 [33] | DWT, PCA | SVM | Bonn University, 2 classes (O+ Z, S) | 98.67% |
| Jose et al., 2020 [18] | Energy, PSD, spectral entropy of EEG sub-band | ELM | Bonn University, 2 classes (S, Z) | 98.5% |
| Sarić et al., 2020 [17] | Time–frequency features of CWT | MLP-ANN | TUH EEG Corpus, 3 classes (FNS, GNSZ, NS) | 95.14% |
| Sahani et al., 2021 [19] | Optimized VMD | Semi-supervised reduced deep CNN (RDCNN) | Bonn University, 2 classes (S, Z) | 99.37% |

## 6. Conclusions

We implemented an epileptic classification system based on the EEG signal using an FPGA. The system was built using the Hjorth descriptor as a feature extraction method and KNN as a classifier. The main digital system architecture to calculate the Hjorth parameter consists of variance, subtractor, divider, and square root. The Hjorth parameter will generate activity, mobility, and complexity for each EEG signal, compiling a set of features. This set becomes the input for the KNN block to be classified as normal, interictal, or ictal. The implemented KNN configuration includes distance calculations using the Euclidean approach and variations in the values of K = 1, 3, 5, 7, and 9. This system is implemented on a Zynq-7000 FPGA device where the EEG signal is sent serially using the UART protocol.

Performance evaluation of the proposed system was carried out on 300 EEG signals with a ratio of training and test data of 64:36, respectively. The constructed system generates the highest accuracy of 90.74% at a K value of 9. From the calculations, the total memory LUT used is below 10%, indicating that the system built is efficient. The processing time used to obtain accurate measurement is only 0.015 s; hence, the proposed method is effective in real time. This study is the basis for designing a system-on-chip (SoC) to detect seizures based on EEG signals by adding a wearable digital EEG device. In the future, this device may be installed in epilepsy patients to provide an alarm when detecting the onset of seizures even before the onset of a seizure. Future research will entail the synthesis of logic circuits that have been designed to become a chip. Another issue with more sophisticated signal processing techniques is the next exciting topic that needs to be investigated to improve accuracy.

**Author Contributions:** Conceptualization, A.R. and S.H.; research methodology, A.R. and S.H.; logic design, A.Z.R.; test bench VHDL and FPGA implementation, A.Z.R. and S.H.; data curation, A.R. and S.H.; writing—original draft preparation, A.R., S.H. and A.Z.R.; writing—review and editing, S.H; visualization, A.R. All authors have read and agreed to the published version of the manuscript.

**Funding:** The APC was funded by Telkom University.

**Data Availability Statement:** Epileptic EEG data were sourced from an open dataset which can be downloaded at https://www.upf.edu/web/ntsa/downloads/ (accessed on 20 April 2022).

**Acknowledgments:** The authors are grateful to the mechatronic workshop and electronic laboratory at Telkom University for supporting this research.

**Conflicts of Interest:** The authors declared no conflict of interest.

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
