# Peer review of "FPGA-Based Implementation for Real-Time Epileptic EEG Classification Using Hjorth Descriptor and KNN"

_electronics, doi:10.3390/electronics11193026_

Round 1
Reviewer 1 Report
The authors did a lot of work, and I believe this is important to the community.
Please compare your work to some of the state of the art techniques to be able to provide a performance baseline.
how your work is different from the work presented in the
https://www.mdpi.com/1424-8220/22/5/1852
Author Response
Thank you for your valuable input. We have added a new paragraph in the Introduction explaining the latest research on real-time epileptic EEG detection using FPGA.
“One application of real-time EEG signal processing is for the detection of epilepsy. Saric et al. used continuous wavelet transform (CWT) and MLP-ANN to detect epilepsy based on EEG signals [17]. The method was implemented using FFGA and tested using the Temple University Hospital Seizure Detection Corpus (TUH EEG Corpus) database. The highest accuracy reached 95.14% using the configuration (5-12-3) on the MLP-ANN. Meanwhile, Jose et al. used an FPGA implementation of EEG epileptic detection using extreme learning (ELM) [18]. The EEG signal was separated into several bands: gamma, beta, alpha, theta, and delta brain rhythms, furthermore modelled by linear prediction theory. Another method to implement epilepsy detection using FPGA is variational mode decomposition (VMD) [19]. This study used reduced deep convolutional neural networks (RDCNN) and multi-kernel random vector functional link networks (MKRVFLN) as classifiers. In the studies mentioned above, the focus of research tends to be on classification methods with advanced feature extraction methods. Exploration of simple methods without signal transformation has not been done much.”
In addition, we added Table 2 for comparison and discussed it in the discussion section.
“Table 2 presents several studies using FPGAs for the detection of epilepsy. Most used datasets were from Bonn University [21], while others used datasets from Temple University Hospital [34]. Meanwhile, for signal processing, all previous studies used a decomposition process, starting from DWT, CWT, and VMD, and the decomposition of EEG signals into the delta, theta, alpha, beta, and gamma signals. Previous research used various classifiers from SVM and ELM to deep learning. The proposed method generally produces lower accuracy compared to earlier studies in Table 2. However, the method used in this study is relatively simple and does not require decomposition or signal transformation with a classifier that does not require a separate training process. It is hoped that the results of this study can be used for real-time detection of epilepsy in research on EEG signals.”
The difference between this study and https://www.mdpi.com/1424-8220/22/5/1852 is that long-term EEG with multimodal feature extraction in the time domain, including signal complexity and first and second-order statistics, was used in this study. We have added the discussion to the Introduction.
“Another study simulated epilepsy detection on long-term EEG using multimodal feature extraction in time domain including signal complexity and first and second-order statistics [10]. Feature extraction in the time domain using a signal complexity approach is also often performed on multiscale signals as reported in the study [11–13]. Most studies related to simulation of classification or detection of epileptic EEG yield high accuracy.”
Reviewer 2 Report
The authors proposed the FPGA‐Based Implementation for Real‐Time Epileptic EEG Classification Using Hjorth Descriptor and KNN, the real-time epileptic EEG classification system that reduces computational cost and time. The paper is in good shape but needs to improve in many aspects, such as necessary research, previous work, Hjorth Descriptor and KNN etc.
The authors should clarify the following points:
· In the abstract part, Therefore, this research proposed the use of a
· “Digital system on FPGA devices to classify epileptic EEG in real‐time on a processor”. This sentence doesn’t make any sense. What processor? What is the main motive of the research classification or prediction?
· The Introduction parts are not enough to emphasize the proposed method. It fails to explain how the classification algorithms are used for classification or prediction.
· An explanation of the time and frequency of the signal is more than required.
· Why do authors not add recent work related to FPGA‐Based Implementation for Real‐Time Epileptic EEG Classification?
· The proposed system showed better performance in classifying normal, interictal and ictal. Could you explain the performance result of your system for five-class classification in real-time?
· I think the Result and Discussion require considerable improvement. Authors should discuss the results and how they can be interpreted from the perspective of previous studies and the working hypotheses.
· Comparison with the other method and difference from other approach is not shown.
· In the conclusion section authors mention that “This research was conducted using a simple feature extraction method, the Hjorth descriptor”. So overall, the innovation of this study is not apparent. The conclusion part is too short and not well-concluded the article.
· Add the failure cases of the proposed system.
· The references are not in MDPI or MDPI recommended standard format.
The citation is not enough to justify the proposed workThe authors proposed the FPGA‐Based Implementation for Real‐Time Epileptic EEG Classification Using Hjorth Descriptor and KNN, the real-time epileptic EEG classification system that reduces computational cost and time. The paper is in good shape but needs to improve in many aspects, such as necessary research, previous work, Hjorth Descriptor and KNN etc.
The authors should clarify the following points:
· In the abstract part, Therefore, this research proposed the use of a
· “Digital system on FPGA devices to classify epileptic EEG in real‐time on a processor”. This sentence doesn’t make any sense. What processor? What is the main motive of the research classification or prediction?
· The Introduction parts are not enough to emphasize the proposed method. It fails to explain how the classification algorithms are used for classification or prediction.
· An explanation of the time and frequency of the signal is more than required.
· Why do authors not add recent work related to FPGA‐Based Implementation for Real‐Time Epileptic EEG Classification?
· The proposed system showed better performance in classifying normal, interictal and ictal. Could you explain the performance result of your system for five-class classification in real-time?
· I think the Result and Discussion require considerable improvement. Authors should discuss the results and how they can be interpreted from the perspective of previous studies and the working hypotheses.
· Comparison with the other method and difference from other approach is not shown.
· In the conclusion section authors mention that “This research was conducted using a simple feature extraction method, the Hjorth descriptor”. So overall, the innovation of this study is not apparent. The conclusion part is too short and not well-concluded the article.
· Add the failure cases of the proposed system.
· The references are not in MDPI or MDPI recommended standard format.
The citation is not enough to justify the proposed work
Author Response
Response for Reviewer 2 is attached in the word file.
Thank you

Reviewer 3 Report
The paper presents a method for real-time classification of epileptic EEG signals using the Hjorth descriptor for feature extraction and the KNN algorithm for normal, ictal and interictal signal classification implemented on an FPGA.
The topic is very interesting and very relevant as there are many studies that try to detect changes in the EEG signal associated with epileptic seizures. The paper is generally correct and well presented. However, I have a few comments for the authors.
- The Introduction section lacks a greater number of papers mentioning the different approaches being used to try to detect these epilectic attacks, from a general or more classical point of view (design of systems for EEG measurement in epilectic people), to works with machine learning solutions such as the presented here.
- The authors cite the database used in the paper. However, the link to the download website is either incorrect or needs to be updated, as it did not allow me to access it to check the description.
- In section 2.2 (Feature Extraction) the authors present the parameters associated with the Hjorth descriptor and the equations that define them. However, they only mention the meaning of the "activity" parameter and what it represents, but not the "mobility" and "complexity" parameters. Furthermore, the equations are presented but the variables associated with them are not described. Subsequently, in Section 3.1 (Feature Extraction Datapath Architecture), the authors present the parameters again and this time they do mention their physical meaning in the signal.
- Improve the description of the captions of the figures. Most of them are very simple and some do not make it clear what the figure refers to.
- Add description of the presented formulas and their parameters, as only in some of them it is done.
Author Response
The paper presents a method for real-time classification of epileptic EEG signals using the Hjorth descriptor for feature extraction and the KNN algorithm for normal, ictal and interictal signal classification implemented on an FPGA.
The topic is very interesting and very relevant as there are many studies that try to detect changes in the EEG signal associated with epileptic seizures. The paper is generally correct and well presented. However, I have a few comments for the authors.
- The Introduction section lacks a greater number of papers mentioning the different approaches being used to try to detect these epilectic attacks, from a general or more classical point of view (design of systems for EEG measurement in epilectic people), to works with machine learning solutions such as the presented here.
Response:
Thank you for the advice. We have added a few sentences and a new paragraph to the Introduction to show previous research studies and the reasons for using KNN as a classifier in this study.
Sentences in Introduction:
“Another study simulated epilepsy detection on long-term EEG using multimodal feature extraction in time domain including signal complexity and first and second-order statistics [10]. Feature extraction in the time domain using a signal complexity approach is also often performed on multiscale signals as reported in the study [11–13]. Most studies related to simulation of classification or detection of epileptic EEG yield high accuracy. However, the various techniques developed have only been realized in the form of software.”
A new paragraph in Introduction:
“One application of real-time EEG signal processing is for the detection of epilepsy. Saric et al. used continuous wavelet transform (CWT) and MLP-ANN to detect epilepsy based on EEG signals [17]. The method was implemented using FFGA and tested using the Temple University Hospital Seizure Detection Corpus (TUH EEG Corpus) database. The highest accuracy reached 95.14% using the configuration (5-12-3) on the MLP-ANN. Meanwhile, Jose et al. used an FPGA implementation of EEG epileptic detection using extreme learning (ELM) [18]. The EEG signal was separated into several bands: gamma, beta, alpha, theta, and delta brain rhythms, furthermore modelled by linear prediction theory. Another method to implement epilepsy detection using FPGA is variational mode decomposition (VMD) [19]. This study used reduced deep convolutional neural networks (RDCNN) and multi-kernel random vector functional link networks (MKRVFLN) as classifiers. In the studies mentioned above, the focus of research tends to be on classification methods with advanced feature extraction methods. Exploration of simple methods without signal transformation has not been done much.”
Several sentences why we choose KNN
For classification, we used K-NN as a classifier. K-NN was chosen because no training process is needed to build the model as in other classifiers. K-NN only requires training data to calculate the distance from the test data. Thus, the implementation process becomes more straightforward.
- The authors cite the database used in the paper. However, the link to the download website is either incorrect or needs to be updated, as it did not allow me to access it to check the description.
Response: we have revised the link of the database: https://www.upf.edu/web/ntsa/downloads/
- In section 2.2 (Feature Extraction) the authors present the parameters associated with the Hjorth descriptor and the equations that define them. However, they only mention the meaning of the "activity" parameter and what it represents, but not the "mobility" and "complexity" parameters. Furthermore, the equations are presented but the variables associated with them are not described. Subsequently, in Section 3.1 (Feature Extraction Datapath Architecture), the authors present the parameters again and this time they do mention their physical meaning in the signal.
Response: We have added additional explanation about Hjorth descriptor in Section 2.2
"The activity defined as the variance of time function , which can be described as square of average distance from each sample to its mean.
The mobility parameter represents the proportion of standard deviation of the power spectrum. This is defined as the division result of variance of the first derivative of the signal x(n) and variance of the signal x(n). Lastly, the complexity parameter represents the change in frequency which calculated by dividing the standard deviation comparison between second to first derivative, and first derivative to the original signal."
And we have added equation how we implement it in Section 3.1
"Therefore, to implement the algorithm, the statistical parameters need to be calculated as running mean and variance using Equation 4."
- Improve the description of the captions of the figures. Most of them are very simple and some do not make it clear what the figure refers to.
Response: We have improved the caption in the figures to make it clearer.
- Add description of the presented formulas and their parameters, as only in some of them it is done.
Response: we have added some explanation about the formula
Round 2
Reviewer 3 Report
In response to the demands of the previous review, authors have added new information to complete the Introduction section as requested in the previous revision. They also updated the link provided to the dataset used in the paper and have added a better description of the Hjorth parameters.
However, the authors comment that they have improved the captions of the figures and all of them remain the same as in the previous version of the paper. I consider that some of them need to be more explanatory.
Author Response
We apologize for our error in understanding the request from the reviewer in the previous round.
We have provided a more complete explanation of all the information from the image according to the reviewer's request. All changes have been highlighted from the caption of figures 1 - 12.
